# Adherence to the CDK 4/6 Inhibitor Palbociclib and Omission of Dose Management Supported by Pharmacometric Modelling as Part of the OpTAT Study

**DOI:** 10.3390/cancers15010316

**Published:** 2023-01-03

**Authors:** Carole Bandiera, Isabella Locatelli, Perrine Courlet, Evelina Cardoso, Khalil Zaman, Athina Stravodimou, Ana Dolcan, Apostolos Sarivalasis, Jean-Philippe Zurcher, Veronica Aedo-Lopez, Jennifer Dotta-Celio, Solange Peters, Monia Guidi, Anna Dorothea Wagner, Chantal Csajka, Marie P. Schneider

**Affiliations:** 1School of Pharmaceutical Sciences, University of Geneva, 1211 Geneva, Switzerland; 2Institute of Pharmaceutical Sciences of Western Switzerland, University of Geneva, 1211 Geneva, Switzerland; 3Center for Primary Care and Public Health (Unisanté), University of Lausanne, 1011 Lausanne, Switzerland; 4Center for Research and Innovation in Clinical Pharmaceutical Sciences, Lausanne University Hospital, University of Lausanne, 1011 Lausanne, Switzerland; 5Precision Oncology Centre, Department of Oncology, Lausanne University Hospital, University of Lausanne, 1011 Lausanne, Switzerland; 6Department of Oncology, Lausanne University Hospital, University of Lausanne, 1011 Lausanne, Switzerland; 7Service of Clinical Pharmacology, Lausanne University Hospital, University of Lausanne, 1011 Lausanne, Switzerland

**Keywords:** CDK4/6 inhibitor, palbociclib, neutropenia, medication adherence, electronic adherence monitoring, motivational interviewing, oral anticancer therapy, pharmacists, interprofessionality

## Abstract

**Simple Summary:**

Medication adherence to CDK4/6 inhibitors such as palbociclib, prescribed as a cyclic oral anticancer therapy in women diagnosed with advanced breast cancer, may be suboptimal. We evaluated adherence to palbociclib and its impact on pharmacokinetic and pharmacodynamic (PK-PD) profiles. Patients included in the OpTAT study used an electronic monitor to register each drug intake event and were randomized into the intervention (i.e., interprofessional medication adherence program) or control group (i.e., usual care). Patients in the intervention group (n = 19) had a higher and more stable adherence than control patients (n = 19). The intervention had a larger effect on patients older than 65 and in patients with longer treatment and disease experience. The PK-PD analysis showed that catching up on a missed dose at the end of the cycle increases the risk of severe neutropenia in the next cycle. The interprofessional healthcare team should closely monitor patients’ cycle management to improve prescriptions and decrease toxicity.

**Abstract:**

The cyclin-dependent kinase 4/6 inhibitor (CDK4/6i) palbociclib is administered orally and cyclically, causing medication adherence challenges. We evaluated components of adherence to palbociclib, its relationship with pharmacokinetics (PK), and drug-induced neutropenia. Patients with metastatic breast cancer (MBC) receiving palbociclib, delivered in electronic monitors (EM), were randomized 1:1 to an intervention and a control group. The intervention was a 12-month interprofessional medication adherence program (IMAP) along with monthly motivational interviews by a pharmacist. Implementation adherence was compared between groups using generalized estimating equation models, in which covariates were included. Model-based palbociclib PK and neutrophil profiles were simulated under real-life implementation scenarios: (1) optimal, (2) 2 doses omitted and caught up at cycle end. At 6 months, implementation was slightly higher and more stable in the intervention (n = 19) than in the control (n = 19) group, 99.2% and 97.3% (Δ1.95%, 95% CI 1.1–2.9%), respectively. The impact of the intervention was larger in patients diagnosed with MBC for >2 years (Δ3.6%, 95% CI 2.1–5.4%), patients who received >4 cycles before inclusion (Δ3.1%, 95% CI 1.7–4.8%) and patients >65 (Δ2.3%, 95% CI 0.8–3.6%). Simulations showed that 25% of patients had neutropenia grade ≥3 during the next cycle in scenario 1 versus 30% in scenario 2. Education and monitoring of patient CDK4/6i cycle management and adherence along with therapeutic drug monitoring can help clinicians improve prescription and decrease toxicity.

## 1. Introduction

### 1.1. Background

With 2.3 million new cases and 685,000 deaths in 2020, breast cancer (BC) is the most frequently diagnosed cancer worldwide [1]. In Switzerland, 6300 female patients are diagnosed with BC, and 1400 die from this disease each year [2]. Despite an increased incidence in the last 30 years, mortality has decreased partly due to systemic therapies, including new therapeutic agents such as oral anticancer therapies (OAT) [3]. Although potentially more convenient for patients thanks to self-management [4], the absence of direct medical supervision raises concerns about medication adherence and adverse effect management [5]. Medication adherence is characterized by initiation (i.e., the patient takes the first dose), implementation (i.e., the extent to which a patient takes the medicine as prescribed) and discontinuation (i.e., the patient stops taking the treatment earlier than prescribed) [6]. Persistence of treatment refers to the time between initiation and discontinuation [6]. A recent systematic review showed that adherence to various OAT varies from 23% to almost 100% [7]. This wide range of estimation is mainly due to the variety of methods to measure medication adherence, from patient self-report to pill-count and electronic monitoring, which makes comparison of results difficult. It has been shown that suboptimal adherence influences measured drug plasma concentrations (i.e., pharmacokinetic (PK) profiles) and is significantly associated with poor clinical outcomes [8,9,10].

It is estimated that 30–60% of BC patients are nonadherent to endocrine therapies, which has a direct impact on their survival [11,12]. Resistance to endocrine therapies has led to the development of new therapeutic options, such as cyclin-dependent kinase 4 and 6 inhibitors (CDK4/6i). Palbociclib and ribociclib are two examples of this newer drug class. Approved in 2015, palbociclib became the most common new CDK4/6i prescribed in patients with endocrine-sensitive advanced or metastatic breast cancer (MBC) and human epidermal growth factor receptor 2-negative (HR+/HER2−) tumors, in combination with aromatase inhibitors, tamoxifen or fulvestrant [13,14]. The PALOMA-1/2/3 and Monaleesa-2/3/7 studies showed improved progression-free survival with palbociclib and ribociclib, respectively, prescribed in combination with endocrine therapies [15,16,17,18,19,20]. Palbociclib and ribociclib medication management is a challenge for patients and providers as these two drugs are administered in a cycle mode: once daily for 21 consecutive days (= ON phase) followed by 7 days of treatment break (= OFF phase) [14,21]. In addition, neutropenia is the most commonly reported adverse effect, resulting in frequent dose reductions and extension of OFF-treatment periods to avoid severe neutropenia [22,23].

In this context, we initiated the randomized controlled trial “Optimizing Targeted Anticancer Therapies” (OpTAT) at the Center for Primary Care and Public Health Unisanté and the Lausanne University Hospital (CHUV), Lausanne, Switzerland [24], to evaluate patient adherence to OAT and its relationship with PK and pharmacodynamics (PD).

### 1.2. Objectives

The first objective of this study was to evaluate and compare adherence to palbociclib, a cyclic CDK4/6i, with a special focus on implementation and 12-month persistence, in patients included in the intervention group versus those in the control group. The second objective was to simulate palbociclib PK and neutrophil profiles to describe the impact of adherence on drug concentrations and neutropenia.

### 1.3. Outcomes

The main outcome is medication implementation, described as a binary variable measured for each patient each day over time (1 = the observed number of electronic monitor (EM) openings is exactly as expected; 0 = the number of observed EM openings diverges from the expected number of EM openings). Medication implementation is defined, at each day t, as the proportion of patients correctly taking the medication (proportion of outcome = 1) among patients still under observation that day.

### 1.4. Hypothesis

We hypothesized that patients included in the intervention group would implement and persist better on palbociclib and that suboptimal implementation would influence palbociclib PK (i.e., drug exposure) and the risk of neutropenia.

## 2. Methods

### 2.1. Ethical Considerations and Guidelines

The OpTAT study was approved by the local ethics committee (Vaud, Switzerland) in 2015 (ID 65/15) and was conducted in accordance with the Declaration of Helsinki principles. We reported findings following the ESPACOMP Medication Adherence Reporting Guideline (EMERGE) [25] and the Consolidated Standards of Reporting Trials (CONSORT) [26].

### 2.2. OpTAT Medication Adherence Study

#### 2.2.1. Study Design and Participants’ Enrolment

The protocol of the OpTAT study has been published elsewhere [24]. Briefly, the OpTAT study was divided into two parts: (1) an open 12-month one-centre 1:1 randomized controlled adherence study and (2) an OAT PK-PD relationship study.

Patients were eligible if they were adults with an OAT prescription for solid cancer. Patients were excluded from the adherence study if their treatment was not self-managed (e.g., home caregivers, medical care services or nursing homes, under tutelage) or if they were diagnosed with cognitive disorders. Patients could either participate in the medication adherence study, the PK-PD study or both by signing a specific informed consent form. In total, 130 patients were included in the medication adherence part of the OpTAT study. In this paper, we analysed a subgroup of MBC patients (n = 38) treated with palbociclib.

#### 2.2.2. Procedures for the Medication Adherence Study

Patient randomization was stratified by cancer type and time since OAT initiation (i.e., more or less than 30 days). Patients were randomized after a baseline period of EM use of at least 21 days. The date of the randomization was determined based on the “per-protocol” statement, i.e., the date of the first intervention delivered. Indeed, because of the coronavirus disease COVID-19 pandemic, the first intervention did not always coincide with randomization and could happen some weeks later. Patients who left the study at any moment before randomization were considered control patients (i.e., they used the EM during baseline but did not receive any intervention).

Patients randomized to the intervention group attended the routine Interprofessional Medication Adherence Program (IMAP). It is a theory-based intervention (information-motivation-behavioural skills model of Fisher [27]) developed to support chronically ill patients with their medication management since 1995 at the community pharmacy of Unisanté [28,29,30]. All enrolled patients used the electronic monitor (EM, MEMS and MEMS AS; AARDEX Group, Sion, Switzerland), which registers the date and hour of each EM opening. An LCD screen on the top of the EM informed the patient of the number of EM daily openings. At each monthly 15- to 20 min medication adherence interview, a temporal graph representing the daily medication intake was discussed between the patient and the pharmacist. Based on motivational interviewing, the pharmacist explored the patient’s needs in terms of information, motivation in medication taking and daily behavioural skills (e.g., medication and adverse effect management) [27]. In addition to EM adherence feedback, pharmacy technicians performed pill counts to calculate the difference between the number of pills delivered and the number of pills that should have been taken by the patient during each intervisit monitored period [31]. To promote interprofessionality, the pharmacist emailed a report on the intervention to the patient’s oncologist and health care providers (HCPs) after each visit [24].

Patients randomized in the control group used the EM but did not have any feedback on their medication intake. Control patients visited the pharmacy only for EM refill. EM data were blinded to patients, clinicians, including the pharmacy team and researchers. Pharmacy technicians only counted the number of pills returned to the pharmacy at each intervisit monitored period so that the concordance of treatment implementation between pill count and EM adherence feedback could be verified at the end of the study.

At each visit, patients in both groups were asked to state any EM deviation use during the last monitored period without (in control patients) and before (in intervention patients) seeing the temporal graph representing medication intake (e.g., pocket doses taken outside of the EM for later use, nonmonitored periods in case of hospitalization or EM nouse during holidays) [31]. Deviations were reported in a case report form (CRF).

### 2.3. Database Construction

#### 2.3.1. Patients’ Sociodemographic and Clinical Data

Patient data were extracted from patients’ electronic medical records (Soarian^®^, Cerner) at enrolment and were registered in the secure data depository RedCap (Research Electronic Data Capture, Vanderbilt University) [32]. This includes sociodemographic (i.e., age, civil status, ethnicity) and clinical data (i.e., time since primary BC diagnosis, time since MBC diagnosis, cancer stage, presence of visceral metastases, palbociclib line of treatment for MBC, previous treatments for MBC if any, combined anticancer therapy in addition to palbociclib, number of received palbociclib cycles before inclusion, time since palbociclib initiation, previous oncologic therapies since BC diagnosis, number of oral prescribed chronic nononcologic treatments at inclusion).

#### 2.3.2. EM Database

The number of observed EM daily openings was reconciled rigorously and manually with pill counts and patient reports about EM deviation use. Both the ON and OFF phases were monitored: one EM opening is expected during the ON phase, and no EM opening is expected during the OFF phase. The end date of the monitoring is defined as the date when the EM is returned or the last day of the last 15-day period with an implementation of >50% if EM has not been used in the last period, as previously performed in adherence research [31].

Palbociclib cycle dates were determined based on three different information levels: (1) the electronic medical record reports and the prescription sheet written by the oncologist, (2) the regimen labelled on the EM pill bottle at each pharmacy dispensation and (3) the patient’s report. In the case of discrepancies between these three sources, the cycle dates were established based on the congruency of two sources. Otherwise, the patient’s report was considered the most updated information, as the cycle dates in the electronic medical record are not always updated according to the last blood cell count results.

Main reasons for prescribing treatment transient interruptions or cycle start deferrals (i.e., EM expected opening = 0) were classified as follows: (1) toxicity or adverse effects, (2) alternate treatment (e.g., radiotherapy) and (3) intercurent infection (e.g., SARS-CoV-2). Transient interruptions may also be caused by the healthcare system (e.g., insurance reimbursement confirmation pending) or upon patients’ requests (e.g., the patient asked the oncologist to delay the new cycle start after her holidays).

Implementation was considered optimal (= 1) if the EM daily observed opening was equal to the expected EM opening and was considered suboptimal (= 0) otherwise. Each reason for a suboptimal implementation was coded with a specific code, which allowed us to describe reasons for nonimplementation (e.g., a specific code for a missed tablet and another code in case of double intake of the drug during the day). Codes were also attributed to each reason for the cycle start deferral (e.g., a code in case of neutropenia and another code in case of intercurent infection), transient palbociclib interruptions and discrepancies of cycle dates notification in the electronic medical record.

### 2.4. Statistical Analysis

#### 2.4.1. Descriptive Analysis

Demographic and clinical variables are presented with proportions in both groups; quantitative variables are presented as medians and interquartile ranges (IQR). Clinical, administrative or patients’ personal events that altered the palbociclib cycle dates are described with proportions. Chi-squared or Fisher exact tests were performed to compare the proportions of events that occurred during the palbociclib cycle between groups. A statistically significant test result is considered if *p* < 0.05.

#### 2.4.2. Implementation

The underlying overall implementation was estimated by applying a generalized estimating equation (GEE) model on the daily 0/1 data with an autoregressive correlation structure and a linear and quadratic time effect.

Since all patients started baseline in the control group and some of them moved to the intervention group upon randomization after a varying number of days, the group variable was introduced into the model as two time-dependent variables, representing at each follow-up time t (1) the time spent in the control group and (2) the time spent in the intervention group (linear and quadratic terms). This allowed a different time pattern to be estimated for a patient always staying in the control group or a patient switching to the intervention at a given time point (i.e., at 21 days). The result is summarized by the difference in implementation (95% confidence intervals, 95% CI) between these two hypothetical patients at 6 months, when the intervention is fully effective, and enough patients are still participating.

Patient age, time since MBC diagnosis and number of palbociclib cycles received at inclusion were entered one by one as covariates into the model. The covariates were dichotomized using the median of the variable as the threshold (65 years old, 2-year time span since MBC diagnosis and 4 palbociclib cycles received before inclusion). The results are summarized by the difference in implementation (95% CI) between the intervention and control groups at 6 months for each covariate category.

The effect of the treatment phase (ON or OFF) on implementation is estimated in a separate model, where the phase is represented by two time-dependent variables: time spent in ON and OFF phases, both considered in linear and quadratic forms.

All statistical analyses were performed with the R statistical package [33].

#### 2.4.3. Persistence

Treatment discontinuation was defined based on previously published criteria [34], i.e., when a patient stops the treatment by herself for any reason (e.g., side effects, pill fatigue) or when the treatment is stopped based on a shared patient-oncologist decision because of side effects or other personal reasons. Any other premature treatment stops (e.g., due to cancer progression) or premature cessation of the study (i.e., patient drop-out) were considered censoring times. Persistence is usually analysed by Kaplan–Meier curves.

#### 2.4.4. Pharmacokinetic Modelling

The population PK-PD model previously developed with data from the OpTAT study [35] was used to evaluate the influence of medication implementation on palbociclib PK and neutropenia. We used different medication implementation scenarios based on real-life patients’ implementation schemes retrieved from our database: (A) optimal implementation (i.e., one dose taken during 21 days followed by 7 days of treatment break); (B1) during the 21 days in phase ON, one dose is omitted at day 18, and is not caught up at cycle end (i.e., the OFF phase is initiated for 7 days from day 22); (B2) during the 21 days in phase ON, one dose is omitted at day 18, which is caught up at cycle end at day 22 (i.e., the OFF phase is initiated from day 23 for 6 days, leading to a shorter OFF period before the next cycle start); (C) during the 21 days in phase ON, the 2 first doses are omitted at day 1 and 2, which are caught up at cycle end at day 22 and 23 (i.e., the OFF phase is initiated from day 24 for 5 days, leading to a shorter OFF period before the next cycle start). Based on these implementation scenarios, we simulated palbociclib plasma concentration (ng/mL) and absolute neutrophil count (G/L) over 3 palbociclib cycles in 1000 patients receiving the standard dosage regimen (125 mg once daily for 21 days followed by 7 days of treatment break).

The population median prediction and the 95% prediction intervals (95% PI) for palbociclib plasma concentrations and absolute neutrophil count as a function of time were derived. For visual interpretation of the impact of real-life implementation scenarios on palbociclib PK, we superimposed to the simulated concentration-time profiles the real palbociclib plasma concentrations measured in one patient receiving a standard palbociclib dosage regimen who generally had an optimal palbociclib implementation (scenario A) but who missed a dose at Day 18 once (scenario B).

## 3. Results

### 3.1. Included Patients

The demographic and clinical variables of the 38 patients are presented in Table 1. In the intervention group, face-to-face motivational interviews lasted an average of 18 min; the same additional duration was needed to complete the adherence report for the oncologist. In the control group, the meetings at the pharmacy counter with patients lasted 8 min on average, and the pharmacists spent a median time of 4 min completing the CRF.

Figure 1 describes the flow of patient enrolment and follow-up from inclusion to data analysis. The EM data of one control patient were not analysed, as the patient did not use the EM device. In the intervention and control groups, 12 and 8 participants left the study prematurely due to treatment stopping because of cancer progression or side effects, respectively; 4 patients dropped out in each group.

### 3.2. Palbociclib Implementation and Persistence

In total, 155 cycles were monitored and analysed in the 19 intervention patients and 184 cycles in the 18 control patients.

Figure 2a presents palbociclib implementation estimated by our model for patients staying in the control group after the baseline period of 21 days versus patients switching to the intervention. Two discontinuations due to side effects occurred: one in the intervention group at day 111 and one in the control group at day 355. At 6 months, implementation was slightly higher and more stable in the intervention group than in the control group (99.2% and 97.3%, Δ1.95%, 95% CI 1.1–2.9).

In the control group, patients older than 65 years, patients diagnosed with MBC for more than 2 years and patients who received more than 4 cycles of palbociclib before inclusion had a lower implementation, whereas in the intervention group, these covariates did not impact implementation. Thus, the impact of the intervention on implementation was larger in patients diagnosed with MBC for more than 2 years in the intervention and control groups, 99.5% and 95.9% (Δ3.6%, 95% CI 2.1–5.4%) (Figure 2b), in patients who received more than 4 cycles of palbociclib before inclusion in the intervention and control groups 98.7% and 95.6% (Δ3.1%, 95% CI 1.7–4.8%) (Figure 2c) and in patients older than 65 in the intervention and control groups, 98.8% and 96.5% (Δ2.3%, 95% CI 0.8–3.6%), respectively (Figure 2d). The results are detailed in Appendix A.

The analysis of ON and OFF phases of all participants—regardless of the randomized group–estimated that implementation decreased over time during ON phases but remained stable in OFF phases during the first 6 months (Appendix B). Equations of the GEE models are presented in Appendix C.

Figure 3 describes the reasons for treatment discontinuation and censoring times. As only two discontinuations occurred in our sample (represented in the implementation graph in Figure 2a), we decided not to formally analyse persistence by Kaplan–Meier curves.

### 3.3. Recommendations for CDK4/6i Cycle Management

#### 3.3.1. Missed Dose Management Recommendation Supported by Pharmacometric Modelling

Forgetting a dose was not uncommon in our population. Significantly more control patients versus intervention missed at least one dose during the monitoring (15/18, 83% and 10/19, 53%, *p* = 0.046), and more cycles were impacted by a missed dose in control versus intervention patients (44/184, 24%, 18/155, 12%, *p* = 0.004) (Table 2). In patients who missed at least one dose, a comparable proportion of patients in both groups caught up these missed doses (7/10, 70% in the intervention and 10/15, 67% in the control group, *p* = 1.000). In the case of a missed dose, some patients did not behave consistently; they caught up the missed dose in some cycles but did not in others (patients who behaved inconsistently in case of a missed dose were 3/10, 30% in intervention versus 7/15, 47% in control group, *p* = 0.679). Some OFF phases were shortened to 6 instead of 7 days in the intervention and control groups, respectively, because a dose was caught up (2/11, 18% and 5/15, 33%, *p* = 0.658) (Table 2). Few patients experienced a dose reduction due to neutropenia (3/19, 16% in the intervention and 2/18, 11% in the control group, *p* = 1.000), and none of these dose reductions were caused by a reduced length of the OFF phase due to the catch-up of missed doses.

As a reminder, the simulated scenarios of palbociclib implementation were (A) optimal implementation, (B1) one dose is omitted at day 18, which is not caught up at cycle end, 7 days of treatment break are initiated from day 22, (B2) one dose is omitted at day 18, which is caught up at cycle end at day 22, 6 days of treatment break are initiated from day 23 (C) the 2 first doses are omitted at day 1 and 2 and are caught up at cycle end at day 22 and 23, 5 days of treatment break are initiated from day 24.

Model-based PK-PD simulations showed that missing one (scenarios B1 and B2) or two doses (scenario C) occasionally resulted in a decrease in the palbociclib plasma concentration-time profile compared to optimal implementation (scenario A) immediately after the missed dose(s), as expected. As shown in the real patient palbociclib plasma concentration represented by the blue dot in Figure 4a scenario B1 vs. the green dots in scenario A, the plasma concentration can even drop below the 95% PI after the missed dose at Day 18. However, compared to an optimal implementation (scenario A), missing a dose in a cycle without catching it up at cycle end (Figure 4a, scenario B1) has no consequence on the recovery from neutropenia (Figure 4b, scenario B1). Conversely, omitting a dose and catching it up at the end of the cycle (Figure 4a, scenario B2) shortens the length of the OFF phase, which may decrease the recovery time and, as a consequence, increase the risk of severe neutropenia during the next cycle (Figure 4b, scenario B2). This risk is even more pronounced when the number of doses omitted and caught up increases, as does the catching-up at the end of the cycle of two doses omitted at the beginning of the cycle (scenario C in Figure 4c,d). Indeed, model-based simulations showed that 25% of patients had neutropenia grade ≥3 during the next cycle in scenario A versus 30% in scenario C.

#### 3.3.2. Transient Interruption during the ON Phase, Cycle Deferrals and Discrepancies of Cycle Dates in the Electronic Medical Record and the Prescription Sheet

At least one transient interruption of palbociclib during the ON phase was experienced in 6/19 (32%) patients in the intervention group and 3/18 (17%) in the control group (*p* = 0.447) (Table 2), with more cycles impacted in the intervention group than in the control group (10/155, 6% vs. 4/184, 2%, *p* = 0.049). Half of the interruptions occurred because of an infection. Other reasons reported were surgery, side effects and synchronization with the fulvestrant cycle or synchronization with previous palbociclib cycles. Among the cycles interrupted, the number of cycles that were resumed after the interruption was 4/10 (40%) in the intervention group and 2/4 (50%) in the control group (*p* = 1.000), whereas the other cycles were stopped for at least 7 days before starting another cycle.

Most patients (13/19, 68% in the intervention and 15/18, 83% in the control group, *p* = 0.447) experienced at least one cycle start deferral due to clinical reasons (e.g., neutropenia, radiotherapy sessions, infections), administrative reasons (e.g., medical appointment set too late, insurance decision for reimbursement pending, delayed order of the treatment at the external pharmacy) or patient’s personal reasons (e.g., more than 20% of patients in each group asked their oncologist to delay the start of the new cycle for personal reasons as holidays) (Table 2).

These frequent transient palbociclib interruptions and deferrals of cycle start are probably the cause of discrepancies in the cycle dates notification in patients’ electronic medical record and prescription sheets (Table 2).

## 4. Discussion

### 4.1. Main Results

In the OpTAT study, patients taking OAT for solid cancer were randomized to the Interprofessional Medication Adherence Program (IMAP) versus controls. In this paper, we present the impact of this intervention in the subgroup of patients treated with palbociclib for MBC. The IMAP, offered to patients in the intervention group, allowed maintaining a higher and more stable palbociclib implementation compared to control patients. The intervention had a larger effect in patients older than 65 years, those diagnosed with MBC for more than 2 years and patients who received more than 4 cycles of palbociclib before inclusion. The intervention did not impact persistence to palbociclib. Model-based PK-PD simulations showed that catching up missed doses at the end of the ON phase, followed by a shortening of the OFF phase, leads to a higher risk of severe neutropenia during the following cycle. Our results show that cycles start deferral occurred due to neutropenia, but also administrative reasons or patients’ personal requests. The interprofessional health care team should closely monitor patients’ cycle management to update cycle dates in the prescription sheets and electronic medical record.

### 4.2. Impact of IMAP on Palbociclib Implementation

Overall, patient implementation was high (>95%) at 6 months in both randomized groups, which indicates efficient treatment self-management. Facilitators for adherence may have been the frequent medical appointments (i.e., usually once per month), sometimes coupled with other consultations from the interprofessional health care team (i.e., nurses, clinical pharmacists, psychologists) to help the patient cope with treatment and disease self-management. Such added consultations to the medical appointments are not offered systematically but according to the patient’s request and needs. Supporting medication adherence is not the core of these interventions and is not embedded in any theoretical framework. Interestingly, our population is not polymedicated (i.e., less than 5 chronic nononcologic treatments are prescribed per day) [37], which may be a facilitator for treatment adherence compared to polymedicated patients. The largest difference in implementation between groups was found between patients diagnosed with MBC for more than 2 years (i.e., +3.6% at 6 months in the intervention group). We hypothesize that patients at a longer distance from the diagnosis may benefit more from the intervention, whereas the usual support from the healthcare team has decreased. A longer time since diagnosis has previously been reported as a determinant of nonadherence to OAT [5,38,39], which may be caused by treatment fatigue [40], a lower perceived need for OAT over time and a lower perceived susceptibility to the disease. Furthermore, the impact of +3.6% of increased implementation on clinical outcomes, such as progression-free survival, needs to be further investigated.

### 4.3. Patient Empowerment to Self-Manage CDK4/6i

Even if patients with BC describe OAT as convenient to use compared to intravenous chemotherapy [41], the numerous dose adjustments, cycle deferrals or interruptions represent a treatment burden, which may impact patient adherence. To better understand patients’ cycle management and avoid any confusion in cycle dates, strategies must be reinforced in daily practice to collect the accurate dates of each cycle start and stop in the electronic medical record, the prescription sheet and on the patient educational written or e-documentation (e.g., “Take one pill per day from 1st January 2022 to 20th January 2022 included, followed by 7 days of treatment break”). In this regard, patients acknowledge the role of pharmacists in providing them education about the treatment [42].

In our study, various behaviours were observed in the same patients after a missed dose, which highlights that consensual education provided by HCPs to self-manage a missed dose might not be sufficient. If oncologists are not aware of patient missed doses and end-of-cycle catching-up behaviours, a low neutrophil blood count can be misinterpreted, leading (i) to unnecessary dose reductions, (ii) treatment interruptions, (iii) next cycle start deferral, and so on to desynchronization with concomitant cyclic treatments, e.g., fulvestrant injections. Interprofessional partnerships are to be strengthened and better defined to synergistically improve patient adherence to palbociclib.

### 4.4. Electronic Adherence Monitoring Databases for Cyclic Regimens

In this study, implementation was monitored in OFF phases (i.e., daily implementation = 1 when observed EM opening = 0), even though the patient’s effort to reach an optimal implementation during the ON phase (i.e., the patient takes a pill once per day) is higher than in the OFF phase. Indeed, we showed that implementation during the ON phases decreased with increasing cycles, whereas it was stable in the OFF phases. Notably, extended OFF phases (e.g., during several weeks for radiotherapy sessions) must be considered differently (e.g., intermittent discontinuation). Further scientific methodologies to characterize the implementation of treatment with cyclic regimens must be investigated (e.g., based on the number of completed cycles over the year with minimal regimen deviation). Data cleaning is of particular importance when numerous prescribed treatment interruptions are monitored to avoid misinterpreting EM nonopenings. As automated and robust methodologies to clean EM databases are needed, we are currently developing a script in the statistical software R to clean and reconcile EM data according to patient’s report and patient’s electronic medical record.

Finally, some patients actively participate in shared decision-making about their treatment management (e.g., asking clinicians for cycle deferral during their holidays). The way to consider them needs to be defined before analysing medication implementation (and adherence) either by considering it as a treatment nonimplementation as it deviates from the recommended regimen or as an optimal treatment implementation as the new regimen relies on a shared-decision process.

### 4.5. Strengths and Limitations

This study has several strengths. First, medication adherence was evaluated by EM, which is considered one of the gold standard methods to measure adherence and especially implementation [43,44,45]. Although patients feeling observed may improve their adherence, this Hawthorn effect fades away within 40 days of monitoring [46,47,48,49]. Second, EM data were reconciled with pill counts [44,49,50]. Third, the database was cleaned rigorously, and the cycle dates were reconciled based on patient reports, medical and pharmaceutical records and prescription sheets. Advanced longitudinal statistical analysis including covariates was performed, which allows estimating patterns of adherence over time [34] by considering the impact of covariates on the implementation trajectories. Last, to the best of our knowledge, the catching-up behaviour was not previously reported in patients taking palbociclib, and the consequences on PK and PD properties are not investigated yet. Indeed, while official instructions for palbociclib stipulate not to catch up a missed dose during the same day, no information is provided on the possibility to catch up a missed dose at cycle end [51].

Some limitations should be acknowledged. First, palbociclib implementation in control patients seems to increase towards the study end. This phenomenon was already observed in a study depicting patient implementation to antiretrovirals in patients with human immunodeficiency virus (HIV) [52] and was defined as a data artefact associated with the decreased number of patients over time. Second, our sample size (n = 38) was relatively small. However, this subgroup represents the largest proportion of patients treated with the same OAT (i.e., palbociclib) among all patients included in the OpTAT study (n = 38/130, 29% of the total sample size). Our robust methodology (i.e., 12-month longitudinal repeated, real-time measures) allows us to provide a solid internal validity of the results. Yet, the external validity needs confirmation with larger multi-center studies. Last, the control group may have been polluted by the implementation of the intervention group at the pharmacy: (i) the exact cycle dates were written in the EM labels at study initiation in both groups, which is not frequent in standard of care if the cycle dates are not written in the prescription sheet; (ii) during the implementation of the OpTAT study at the pharmacy, community pharmacists have improved their scientific knowledge and expertise about OAT, which might have influenced the professional attitude with the control patients; (iii) for ethical reasons, patients in both groups were recalled if they did not show up after a missed appointment for the intervention or a missed date for refill in control patients, respectively, in the next 24 h for intervention patients and after 72 h for control patients, which does not happen in usual care; (iv) participation in the OpTAT study and EM use might have raised awareness of the importance of medication adherence in control patients; (v) to avoid EM misuse, control patients were asked to report their deviation of EM use at each refill visit.

## 5. Conclusions

The interprofessional medication adherence program shows that the intervention had a larger effect in patients older than 65 and those with longer treatment and disease experience. Various clinical, administrative or patients’ personal events can alter the dates of ON and OFF phases. OFF-period reduction due to missed doses that are caught up at cycle end increases the risk of severe neutropenia in the next cycle, which may lead to inappropriate dose reduction. Currently, two CDK4/6i are administered cyclically, palbociclib and ribociclib. Well-designed education programs and monitoring of patient cycle management by an interprofessional team along with PK-piloted dose reduction in case of side effects can help clinicians improve prescriptions and better prevent toxicity. The impact of adherence on CDK4/6i effectiveness should be further investigated.

## Figures and Tables

**Figure 1 cancers-15-00316-f001:**
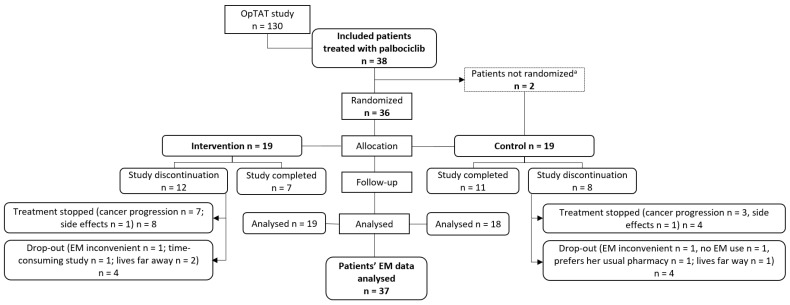
Flow of patient enrollment and follow up from inclusion to data analysis. ^a^ study discontinuation before randomization; NB: EM = electronic monitor.

**Figure 2 cancers-15-00316-f002:**
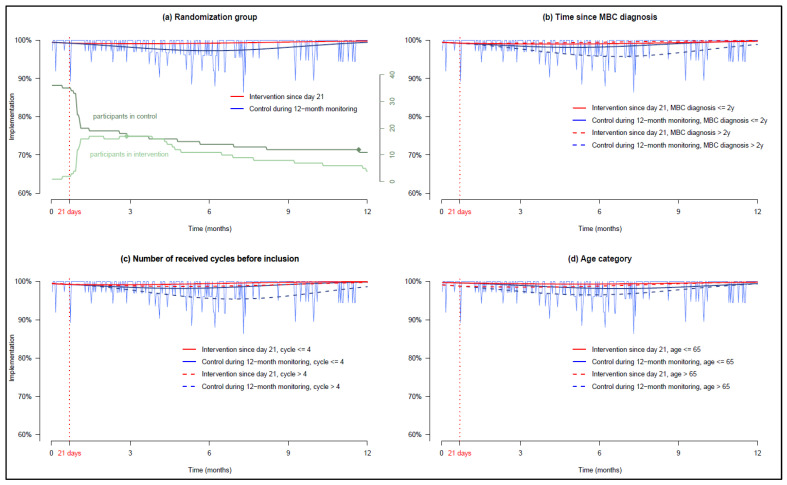
GEE models estimating patient implementation (the vertical red dotted line indicates the randomization at day 21, date on which the intervention starts—i.e., after the baseline period). (**a**) Patients’ palbociclib implementation in each randomized group, (**b**) dichotomization in each group according to time since MBC diagnosis, (**c**) dichotomization in each group according to the number of palbociclib cycle received before inclusion, and (**d**) dichotomization in each group according to age. NB: In (**a**), the two discontinuations are represented by dots in the green curves depicting the number of participants over time.

**Figure 3 cancers-15-00316-f003:**
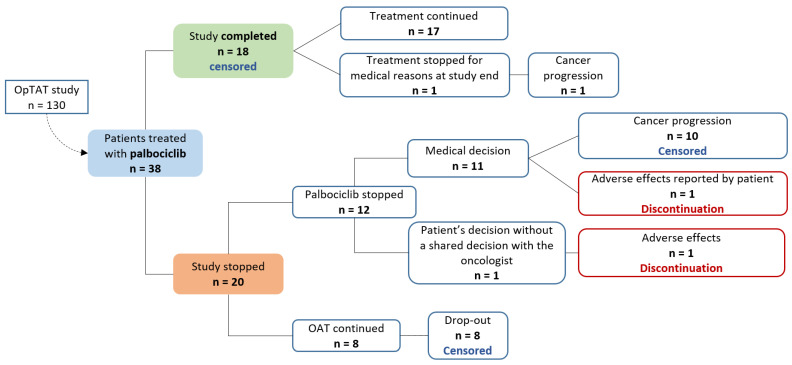
Discontinuation and censoring times algorithm.

**Figure 4 cancers-15-00316-f004:**
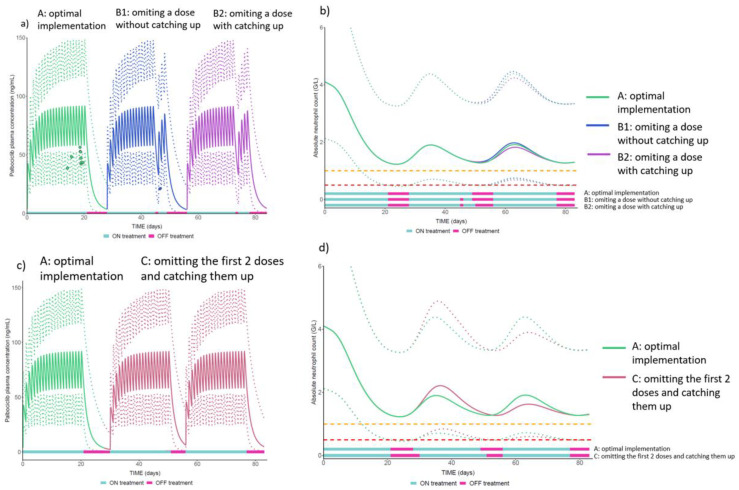
Population median prediction of palbociclib plasma concentrations profiles–solid lines in (**a**,**c**)—and absolute neutrophil count–solid lines in (**b**,**d**)—and their 95% prediction intervals–dotted lines. Orange and red dotted lines represent, respectively the threshold for neutropenia grade ≥3 and ≥4. NB: the dots in (**a**) represent the observed plasma palbociclib concentrations in a real patient included in the OpTAT study in both A and B1 scenarios. In scenario A, the concentrations are lower than the median observed in the simulated patients. This observation is consistent over this patient’s cycles and could be attributed to several extrinsic factors potentially increasing palbociclib elimination for this patient (e.g., co-administration of proton-pump inhibitors in fasting conditions [35], increase in creatinine clearance or decrease in alkaline phosphatase [36]).

**Table 1 cancers-15-00316-t001:** Demographic and clinical data of patients treated with palbociclib in the OpTAT study (n = 38).

	Intervention (n = 19)	Control (n = 17) + Not Randomized ^c^ (n = 2)
Demographic Data		
Age (years), median (IQR)	62 (52–73)	64 (55–75)
Marital civil status ^a^, n (%)	10 (52.6)	11 (57.9)
Caucasian ethnicity ^b^, n (%)	19 (100)	17 (89.5)
Clinical data		
Time since primary BC diagnosis (years), median (IQR)	7.5 (4.5–16.9)	7.3 (3.1–12.8)
Time since MBC diagnosis (years), median (IQR)	1.8 (0.8–3.0)	1.6 (0.5–3.4)
Cancer stage IV, n (%)	19 (100)	19 (100)
Visceral metastases, n (%)	16 (84.2)	10 (52.6)
Palbociclib line of treatment for MBC, n (%)	1st line: 6 (31.6) 2nd line: 3 (15.8) >= 3rd line: 10 (52.6)	1st line: 5 (26.3) 2nd line: 6 (31.6) >= 3rd line: 8 (42.1)
Previous treatment for MBC (if palbociclib is ≥ 2nd treatment for MBC), n (%)	Endocrine therapy: 11 (84.6) Chemotherapy: 2 (15.4)	Endocrine therapy: 7 (50) Chemotherapy: 7 (50)
Combined anticancer therapy in addition to palbociclib at inclusion, n (%)	Aromatase inhibitor: 7 (36.8) Fulvestrant: 11 (57.9) Goserelin, leuprorelin: 2 (10.5)	Aromatase inhibitor: 6 (31.6) Fulvestrant: 13 (68.4) Goserelin, leuprorelin: 2 (10.5)
Number of palbociclib cycles received before inclusion, n (%)	0–4 cycle(s): 11 (57.9) >4 cycles: 8 (42.1)	0–4 cycle(s): 10 (52.6) >4 cycles: 9 (47.4)
Time since palbociclib initiation (days), median (IQR)	97 (14–230)	83 (28–228)
Previous oncologic therapies since BC diagnosis, n (%)	Tumor surgery: 19 (100) Aromatase inhibitor: 17 (89.5) IV chemotherapy: 14 (73.7) Radiotherapy: 11 (57.9) Fulvestrant: 4 (21.0) Goserelin, leuprorelin: 4 (21.0) Trastuzumab: 0 (0) Bevacizumab: 4 (21.0) Everolimus: 3 (15.8) Capecitabine: 2 (10.5)	Tumor surgery: 19 (100) Aromatase inhibitor: 17 (89.5) IV chemotherapy: 12 (63.2) Radiotherapy: 15 (79.0) Fulvestrant: 3 (15.8) Goserelin, leuprorelin: 1 (5.3) Trastuzumab: 1 (5.3) Bevacizumab: 5 (26.3) Everolimus: 2 (10.5) Capecitabine: 3 (15.8)
Number of oral prescribed chronic nononcologic treatments at inclusion time, median (IQR)	3 (1–4)	3 (1–3)
Adherence study		
Time spent in the adherence study (days), median (IQR)	209 (133–363)	366 (171–392)

NB: IQR = Interquartile range; BC = breast cancer, MBC = metastatic breast cancer; IV = intravenous. ^a^ The other patients are separated, divorced, widowed or single. ^b^ The other patients are African or Hispanic. ^c^ Patients not randomized left the study before randomization.

**Table 2 cancers-15-00316-t002:** Patients’ palbocilib management behaviour; clinical, administrative or patients’ personal events that occurred during the monitored period.

	Intervention Group (n_patients_ = 19; n_cycles_ = 155)	Control Group (n_patients_ = 18; n_cycles_ = 184)	*p*-Value
**Patients’ Behaviour When a Dose is Missed**			
Patients who missed at least one dose, n (%)	10/19 (53)	15/18 (83)	**0.046**
Number of cycles impacted by a missed dose (%)	18/155 (12)	44/184 (24)	**0.004**
Number of ON cycles extended because a missed dose was caught up, n (%)	11/18 (61)	15/44 (34)	0.050
Number of cycles—among those with a caught up dose-in which the OFF phase was shortened from 7 to 6 days, n (%)	2/11 (18)	5/15 (33)	0.658
Number of patients—among those who missed at least one dose-who **caught up** at least one missed dose, n (%)	7/10 (70)	10/15 (67)	1.000
Number of patients—among those who missed at least one dose-who caught up a missed dose **in some cycles and did not caught up in the other cycles**, n (%)	3/10 (30)	7/15 (47)	0.679
**Transient interruptions of palbociblib during the phase ON**			
Number of patients who experienced at least one transient interruption of palbociclib during the ON phase, n (%)	6/19 (32)	3/18 (17)	0.447
Number of cycles impacted by an interruption in phase ON, n (%)	10/155 (6)	4/184 (2)	**0.049**
Number of cycles—among those which were interrupted-that were **resumed** after the interruption, n (%)	4/10 (40)	2/4 (50)	1.000
Number of phases ON—among those with an interruption-interrupted because of infection, n (%)	5/10 (50)	2/4 (50)	1.000
Number of phases ON—among those with an interruption-interrupted because of surgery, n (%)	2/10 (20)	1/4 (25)	1.000
Number of phases ON—among those with an interruption-interrupted because of side effects, n (%)	1/10 (10)	1/4 (25)	0.506
Number of phases ON—among those with an interruption-interrupted because of synchronization with the fulvestrant cycle or with previous palbociclib cycles, n (%)	2/10 (20)	0/4 (0)	1.000
**Cycle start deferrals**			
Number of patients who experienced at least one cycle deferral, n (%)	13/19 (68)	15/18 (83)	0.447
Number of cycles impacted by a deferral, n (%)	40/155 (26)	36/184 (20)	0.170
Number of patients who experienced a cycle deferral due to neutropenia, n (%)	9/19 (47)	7/18 (39)	0.603
Number of cycles deferred because of neutropenia, n (%)	20/155 (13)	11/184 (6)	**0.028**
Number of patients who experienced a cycle deferral due to radiotherapy sessions, n (%)	2/19 (11)	0/18 (0)	0.487
Number of patients who experienced a cycle deferral due to infection, n (%)	3/19 (16)	3/18 (17)	1.000
Number of patients who experienced a cycle deferral due to prevent the risk of SARS-COV-2 infection during the COVID-19 pandemic, n (%)	0/19 (0)	1/18 (6)	0.487
Number of patients who experienced a cycle deferral due to a medical appointment set too late (oncologist not available or PET-scan results pending), n (%)	5/19 (26)	7/18 (39)	0.414
Number of patients who experienced a cycle deferral due to insurance reimbursement decision pending, n (%)	0/19 (0)	1/18 (6)	0.487
Number of patients who experienced a cycle deferral due to the delayed order of the treatment at the external pharmacy, n (%)	1/19 (5)	0/18 (0)	1.000
Number of patients who asked their oncologist to defer the start of at least one cycle for personal reasons (e.g., holidays), n (%)	4/19 (21)	4/18 (22)	1.000
**Dose reduction**			
Patients who experienced a dose reduction due to neutropenia ^a^, n (%)	3/19 (16)	2/18 (11)	1.000
**Discrepancies of cycle dates notification** in the electronic medical record or in the prescription sheet compared to the actual cycles dates			
Number of patients impacted by at **least one discrepancy** in the cycle dates compared to the actual cycle dates, n (%)	10/19 (53)	11/18 (61)	0.603
Number of patients—among those impacted by a discrepancy-for which the prescription was not modified by pharmacists (i.e., the phase OFF was shortened or extended) ^b^, n (%)	4/10 (40)	5/11 (45)	1.000

^a^ The dose was reduced from 100 mg to 75 mg in three intervention patients and one control patient at cycle 5, 10, 20 and 3; as well as from 125 mg to 100 mg in one control patient at cycle 14. This suggests that doses adjustment can occur anytime from treatment initiation. ^b^ the discrepancies of the cycle dates from the electronic medical record was adapted in the prescription sheet by the oncologist, or the prescription sheet was adapted by the pharmacist according to patients’ report and the calculation of cycle dates following the previous cycle.

## Data Availability

The data presented in this study are available on request from the corresponding author. The data are not publicly available due to ethical reasons.

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
