# Peer review of "Adherence to the CDK 4/6 Inhibitor Palbociclib and Omission of Dose Management Supported by Pharmacometric Modelling as Part of the OpTAT Study"

_cancers, 2023, doi:10.3390/cancers15010316_

Round 1

Reviewer 1 Report

CDK4/6i is important for patients with hormone-receptor-positive breast cancer, especially those in the advanced stages. The authors analyzed a subgroup from a large clinical trial and drew conclusions about the possible benefits of their drug intervention. 

This is a very important and interesting study, but I still have some questions/comments:

1) The sample size of the study is relatively small, and solid conclusions may not be drawn.

2) Both the clinical trials and the instructions for palbociclib have mentioned the handling methods once the drug was omitted, which makes the study less innovative.

3)  In addition to the impact on the patient's blood system, we should probably be more concerned about the patient's disease itself. Since this manuscript is from a study that started years, can you show the disease changes in these patients since they were enrolled?

Author Response

We thank you and the reviewer for your comments which helped to enrich the manuscript entitled “Adherence to the CDK 4/6 inhibitor palbociclib and omission of dose management supported by pharmacometric modelling as part of the OpTAT study”. We are pleased to submit the revised manuscript with tracked changes along with point-by-point responses to reviewer’s comments.

Reviewer #1:

Reviewer’s comment

Authors’ response

CDK4/6i is important for patients with hormone-receptor-positive breast cancer, especially those in the advanced stages. The authors analyzed a subgroup from a large clinical trial and drew conclusions about the possible benefits of their drug intervention. 

This is a very important and interesting study, but I still have some questions/comments:

1) The sample size of the study is relatively small, and solid conclusions may not be drawn.

Dear reviewer, thank you for your comments. We agree with you that the sample size is relatively small.

Among the 130 participants included in the OpTAT study, 38 were treated with palbociclib. This subgroup represents the largest proportion of patients treated with the same protein kinase inhibitor (PKI) (i.e., palbociclib) among all patients included in the OpTAT study (n=38/130, 29% of the total sample size).

Monitoring adherence (especially medication implementation) with electronic monitors (EM) is the gold standard, yet it is challenging in clinical practice as it is not yet part of routine care and it requests resources. Based on our review of the literature, we can state that the sample size of most published studies is relatively small. Indeed, even if Partridge et al. included 161 patients[1] and Krolop et al. 73 patients[2] to analyze adherence to capecitabine with EM, most studies included fewer patients: for instance, Mayer and al. included 13 patients treated with gefitinib and capecitabine[3], Linder et al. included 23 adolescents and young adults treated with oral anticancer therapies[4], Thivat et al. included 29 patients treated with capecitabine or lapatinib[5], Figueirudo et al. included 30 patients[6] and Simons et al. 48 patients taking capecitabine[7] and Leader et al. a convenient sample of 58 patients treated with PKI for chronic myeloid leukemia[8].

Moreover, we believe that our robust methodology (i.e., 12-month longitudinal repeated, real-time measures) allows us to provide a solid internal validity of the results. However, we agree that external validity needs to be confirmed with larger multi-center studies. We amended the limitations section of the discussion, see lines 501-506.

2) Both the clinical trials and the instructions for palbociclib have mentioned the handling methods once the drug was omitted, which makes the study less innovative.

Thank you for comment. In the manufacturer’ instructions for palbociclib, it is mentioned “Patients should be encouraged to take their dose at approximately the same time each day. If the patient vomits or misses a dose, an additional dose should not be taken that day. The next prescribed dose should be taken at the usual time.”[9] Two pieces of information are missing: 1) the delay acceptable to catch up a missed dose during the day according to the usual timing of intake (we will investigate this issue in an upcoming paper), 2) there is no information about the expected catching-up/no-catching-up attitude at the end of the cycle, i.e., extending or not the cycle by the number of missed doses during the ON-phase. 

Indeed, in the OpTAT study, oncologists were not aware of the patient catching-up behaviour at the end of the cycle. This behaviour was described by a precise and rigorous analysis of patients’ medication management monitored by EM in routine clinical practice. It is thus not surprising that this behaviour was not addressed in the manufacturer’ instructions. Indeed, as mentioned in the discussion lines 453-455, different behaviours were observed in the same patients after a missed dose, showing that a consensus among healthcare providers on the recommended behaviour after a missed dose is lacking. Thus, we believe that our study is innovative: i) although the 24-hour catching-up behavior is well described in the manufacturer’s documentation, the catching-up behavior at the end of the cycle is not documented at all ; ii) it informs healthcare providers on the need to provide clear recommendations to patients regarding their behaviour at the end of the cycle.

In order to strengthen this point, we added a comment in the strengths and limitations paragraph lines 492-496.

3)  In addition to the impact on the patient's blood system, we should probably be more concerned about the patient's disease itself. Since this manuscript is from a study that started years, can you show the disease changes in these patients since they were enrolled?

Thank you for this valuable comment. In this study, we were not able to analyze the impact of the intervention on clinical outcomes, such as progression-free survival or tumor size. First, numerous covariables affect clinical outcomes, such as the time since breast cancer diagnosis, the time since palbociclib initiation, palbociclib treatment line or concomitant treatments in addition to palbociclib and demographic variables such as age. Thus, it makes it challenging to compare clinical outcomes among patients in the intervention and the control group with a robust methodology with a sample size of 38 patients.

Second, based on the protocol approved by the ethics committee, we were not allowed to investigate clinical outcomes beyond the 12-month study duration.

Our results give robust, innovative information on important surrogate endpoints such as medication adherence, pharmacokinetics and pharmacodynamics of palbociclib in advanced breast cancer. Further larger, multi-center studies are needed to explore the impact on clinical outcomes such as survival and tumor size. This point was addressed in the paragraph entitled “Impact of IMAP on palbocilib implementation” in the discussion section lines 440-442.

We have uploaded the revised manuscript in the submission portal.
Thank you for your consideration,

Sincerely, 

Carole Bandiera and Pr. Marie P. Schneider

References

  1. Partridge, A.H., L. Archer, A.B. Kornblith, J. Gralow, D. Grenier, E. Perez, A.C. Wolff, X. Wang, H. Kastrissios, D. Berry, C. Hudis, E. Winer, and H. Muss, Adherence and persistence with oral adjuvant chemotherapy in older women with early-stage breast cancer in CALGB 49907: adherence companion study 60104. J Clin Oncol, 2010. 28(14): p. 2418-22.
  2. Krolop, L., Y.D. Ko, P.F. Schwindt, C. Schumacher, R. Fimmers, and U. Jaehde, Adherence management for patients with cancer taking capecitabine: a prospective two-arm cohort study. BMJ Open, 2013. 3(7).
  3. Mayer, E.L., A.H. Partridge, L.N. Harris, R.S. Gelman, S.T. Schumer, H.J. Burstein, and E.P. Winer, Tolerability of and adherence to combination oral therapy with gefitinib and capecitabine in metastatic breast cancer. Breast Cancer Res Treat, 2009. 117(3): p. 615-23.
  4. Linder, L.A., Y.P. Wu, C.F. Macpherson, B. Fowler, A. Wilson, Y. Jo, S.H. Jung, B. Parsons, and R. Johnson, Oral Medication Adherence Among Adolescents and Young Adults with Cancer Before and Following Use of a Smartphone-Based Medication Reminder App. J Adolesc Young Adult Oncol, 2019. 8(2): p. 122-130.
  5. Thivat, E., I. Van Praagh, A. Belliere, M.A. Mouret-Reynier, F. Kwiatkowski, X. Durando, H. Mahammedi, A.F. Dillies, P. Chollet, and R. Chevrier, Adherence with oral oncologic treatment in cancer patients: interest of an adherence score of all dosing errors. Oncology, 2013. 84(2): p. 67-74.
  6. Figueiredo Junior, A.G. and N.M. Forones, Study on adherence to capecitabine among patients with colorectal cancer and metastatic breast cancer. Arq Gastroenterol, 2014. 51(3): p. 186-91.
  7. Simons, S., S. Ringsdorf, M. Braun, U.J. Mey, P.F. Schwindt, Y.D. Ko, I. Schmidt-Wolf, W. Kuhn, and U. Jaehde, Enhancing adherence to capecitabine chemotherapy by means of multidisciplinary pharmaceutical care. Support Care Cancer, 2011. 19(7): p. 1009-18.
  8. Leader, A., N. Benyamini, A. Gafter-Gvili, J. Dreyer, B. Calvarysky, A. Amitai, O. Yarchovsky-Dolberg, G. Sharf, E. Tousset, O. Caspi, M. Ellis, I. Levi, S. De Geest, and P. Raanani, Effect of Adherence-enhancing Interventions on Adherence to Tyrosine Kinase Inhibitor Treatment in Chronic Myeloid Leukemia (TAKE-IT): A Quasi-experimental Pre-Post Intervention Multicenter Pilot Study. Clin Lymphoma Myeloma Leuk, 2018. 18(11): p. e449-e461.
  9. EuropeanMedicineAgency. Ibrance : EPAR - Product Information - SUMMARY OF PRODUCT CHARACTERISTICS. 2022 [cited 2022 23.12.2022]; Available from: https://www.ema.europa.eu/en/documents/product-information/ibrance-epar-product-information_en.pdf.

Reviewer 2 Report

Schneider et al., in this manuscript, described Adherence to the CDK 4/6 inhibitor Palbociclib and omission of dose management supported by pharmacometric modeling as part of the OpTAT (Optimizing Targeted Anticancer Therapies) study. Overall, the OpTAT research presented in this manuscript is organized well and meets the Cancers journal standards. Therefore, I recommend publishing this manuscript in its current form.

Author Response

Reviewer’s comment

Authors’ response

Schneider et al., in this manuscript, described Adherence to the CDK 4/6 inhibitor Palbociclib and omission of dose management supported by pharmacometric modeling as part of the OpTAT (Optimizing Targeted Anticancer Therapies) study. Overall, the OpTAT research presented in this manuscript is organized well and meets the Cancers journal standards. Therefore, I recommend publishing this manuscript in its current form.

Dear reviewer, we thank you very much for your positive feedback.

Reviewer 3 Report

This manuscript by Bandiera et al. describes the benefits of an interprofessional medication adherence program in monitoring patient CDK4/6i cycle management. It’s a well-documented OpTAT study showing medication adherence to CDK4/6 inhibitors such as Palbociclib. The experiments adequately support the conclusions, and I recommend accepting the manuscripts in the current format.

Author Response

Reviewer’s comment

Authors’ response

This manuscript by Bandiera et al. describes the benefits of an interprofessional medication adherence program in monitoring patient CDK4/6i cycle management. It’s a well-documented OpTAT study showing medication adherence to CDK4/6 inhibitors such as Palbociclib. The experiments adequately support the conclusions, and I recommend accepting the manuscripts in the current format.

Dear reviewer, we thank you very much for your positive feedback.

Round 2

Reviewer 1 Report

The excellent article highlights the importance of CDK4/6 management, which is currently used more frequently in the clinic for first-line treatment of luminal breast cancer, so this article helps clinicians realize the importance of drug management.

But the number of included patients need be interpreted.

Meanwhile,education programs and monitoring need how much manage costs?

Author Response

                                                                        December 27, 2022

Dear reviewer,

I thank you for your additional comments on our manuscript entitled “Adherence to the CDK 4/6 inhibitor palbociclib and omission of dose management supported by pharmacometric modelling as part of the OpTAT study”. I am pleased to submit the revised manuscript with tracked changes along with point-by-point responses to your comments.

Reviewer #1:

Reviewer’s comment

Authors’ response

The excellent article highlights the importance of CDK4/6 management, which is currently used more frequently in the clinic for first-line treatment of luminal breast cancer, so this article helps clinicians realize the importance of drug management.

But the number of included patients need be interpreted.

Dear reviewer,

We thank you for your additional comments on our paper presenting the importance of CDK4/6i management.

We interpreted the number of included patients in our responses to the first round of review (question 1) and we amended the limitations section of the discussion accordingly, see lines 501-506. If this response appears not sufficient to you, could you please be more specific in your question?

As a reminder, here was our response in the first round of review:

“We agree with you that the sample size is relatively small.

Among the 130 participants included in the OpTAT study, 38 were treated with palbociclib. This subgroup represents the largest proportion of patients treated with the same protein kinase inhibitor (PKI) (i.e., palbociclib) among all patients included in the OpTAT study (n=38/130, 29% of the total sample size).

Monitoring adherence (especially medication implementation) with electronic monitors (EM) is the gold standard, yet it is challenging in clinical practice as it is not yet part of routine care and it requests resources. Based on our review of the literature, we can state that the sample size of most published studies is relatively small. Indeed, even if Partridge et al. included 161 patients[1] and Krolop et al. 73 patients[2] to analyze adherence to capecitabine with EM, most studies included fewer patients: for instance, Mayer and al. included 13 patients treated with gefitinib and capecitabine[3], Linder et al. included 23 adolescents and young adults treated with oral anticancer therapies[4], Thivat et al. included 29 patients treated with capecitabine or lapatinib[5], Figueirudo et al. included 30 patients[6] and Simons et al. 48 patients taking capecitabine[7] and Leader et al. a convenient sample of 58 patients treated with PKI for chronic myeloid leukemia[8].

Moreover, we believe that our robust methodology (i.e., 12-month longitudinal repeated, real-time measures) allows us to provide a solid internal validity of the results. However, we agree that external validity needs to be confirmed with larger multi-center studies. We amended the limitations section of the discussion, see lines 501-506.”

Meanwhile, education programs and monitoring need how much manage costs?

Thank you for your comment. The cost of the interprofessional medication adherence program (IMAP) implemented at the community pharmacy of Unisanté since 1995 was evaluated by a micro-costing analysis and a break-even analysis by Perraudin et al.[9]. This economic evaluation showed that the IMAP allows pharmacists to reach the break-even point (i.e., the required number of patients to follow up with to ensure that the generated revenue exceeds the total cost)[9]. This reference was added in the manuscript, see line 141.

I have uploaded the revised manuscript in the submission portal.
Thank you for your consideration,

Sincerely, 

Carole Bandiera

References

  1. Partridge, A.H., L. Archer, A.B. Kornblith, J. Gralow, D. Grenier, E. Perez, A.C. Wolff, X. Wang, H. Kastrissios, D. Berry, C. Hudis, E. Winer, and H. Muss, Adherence and persistence with oral adjuvant chemotherapy in older women with early-stage breast cancer in CALGB 49907: adherence companion study 60104. J Clin Oncol, 2010. 28(14): p. 2418-22.
  2. Krolop, L., Y.D. Ko, P.F. Schwindt, C. Schumacher, R. Fimmers, and U. Jaehde, Adherence management for patients with cancer taking capecitabine: a prospective two-arm cohort study. BMJ Open, 2013. 3(7).
  3. Mayer, E.L., A.H. Partridge, L.N. Harris, R.S. Gelman, S.T. Schumer, H.J. Burstein, and E.P. Winer, Tolerability of and adherence to combination oral therapy with gefitinib and capecitabine in metastatic breast cancer. Breast Cancer Res Treat, 2009. 117(3): p. 615-23.
  4. Linder, L.A., Y.P. Wu, C.F. Macpherson, B. Fowler, A. Wilson, Y. Jo, S.H. Jung, B. Parsons, and R. Johnson, Oral Medication Adherence Among Adolescents and Young Adults with Cancer Before and Following Use of a Smartphone-Based Medication Reminder App. J Adolesc Young Adult Oncol, 2019. 8(2): p. 122-130.
  5. Thivat, E., I. Van Praagh, A. Belliere, M.A. Mouret-Reynier, F. Kwiatkowski, X. Durando, H. Mahammedi, A.F. Dillies, P. Chollet, and R. Chevrier, Adherence with oral oncologic treatment in cancer patients: interest of an adherence score of all dosing errors. Oncology, 2013. 84(2): p. 67-74.
  6. Figueiredo Junior, A.G. and N.M. Forones, Study on adherence to capecitabine among patients with colorectal cancer and metastatic breast cancer. Arq Gastroenterol, 2014. 51(3): p. 186-91.
  7. Simons, S., S. Ringsdorf, M. Braun, U.J. Mey, P.F. Schwindt, Y.D. Ko, I. Schmidt-Wolf, W. Kuhn, and U. Jaehde, Enhancing adherence to capecitabine chemotherapy by means of multidisciplinary pharmaceutical care. Support Care Cancer, 2011. 19(7): p. 1009-18.
  8. Leader, A., N. Benyamini, A. Gafter-Gvili, J. Dreyer, B. Calvarysky, A. Amitai, O. Yarchovsky-Dolberg, G. Sharf, E. Tousset, O. Caspi, M. Ellis, I. Levi, S. De Geest, and P. Raanani, Effect of Adherence-enhancing Interventions on Adherence to Tyrosine Kinase Inhibitor Treatment in Chronic Myeloid Leukemia (TAKE-IT): A Quasi-experimental Pre-Post Intervention Multicenter Pilot Study. Clin Lymphoma Myeloma Leuk, 2018. 18(11): p. e449-e461.
  9. Perraudin, C., J.-F. Locca, C. Rossier, O. Bugnon, and M.-P. Schneider, Implementation of an interprofessional medication adherence program for chronic patients in community pharmacies: how much does it cost for the provider? BMC Health Services Research, 2019. 19(1): p. 15.

Round 3

Reviewer 1 Report

Thanks very much for your respoend!